# Socio-Economic Factors Associated with Ethnic Disparities in SARS-CoV-2 Infection and Hospitalization

**DOI:** 10.3390/ijerph20156521

**Published:** 2023-08-04

**Authors:** Alessio Gili, Marta Caminiti, Chiara Lupi, Salvatore Zichichi, Ilaria Minicucci, Patrizio Pezzotti, Chiara Primieri, Carla Bietta, Fabrizio Stracci

**Affiliations:** 1Public Health Section, Department of Medicine and Surgery, University of Perugia, 06129 Perugia, Italy; 2Department of Infectious Diseases, Istituto Superiore di Sanità, 00162 Rome, Italy; 3Epidemiology Unit, Department of Preventive Medicine, Local Health Unit 1, 06126 Perugia, Italy

**Keywords:** COVID-19, Italian, non-Italians, Umbria, hospitalization, HDI, deprivation index

## Abstract

Background: There is evidence that non-Italians presented higher incidence of infection and worse health outcomes if compared to native populations in the COVID-19 pandemic. The aim of the study was to compare Italian- and non-Italian-born health outcomes, accounting for socio-economic levels. Methods: We analyzed data relative to 906,463 people in Umbria (Italy) from 21 February 2020 to 31 May 2021. We considered the National Deprivation Index, the Urban–Rural Municipalities Index and the Human Development Index (HDI) of the country of birth. We used a multilevel logistic regression model to explore the influence of these factors on SARS-CoV-2 infection and hospitalization rates. Diagnosis in the 48 h preceding admission was an indicator of late diagnosis among hospitalized cases. Results: Overall, 54,448 persons tested positive (6%), and 9.7% of them were hospitalized. The risk of hospital admission was higher among non-Italians and was inversely related to the HDI of the country of birth. A diagnosis within 48 h before hospitalization was more frequent among non-Italians and correlated to the HDI level. Conclusions: COVID-19 had unequal health outcomes among the population in Umbria. Reduced access to primary care services in the non-Italian group could explain our findings. Policies on immigrants' access to primary healthcare need to be improved.

## 1. Introduction

The COVID-19 pandemic has left no population sector untouched. At the time of this writing, there are 758,900,564 confirmed cases and 6,859,093 deaths worldwide [1]. 

Recent literature suggests that non-Italians are more likely to be infected with SARS-CoV-2 and have worst health outcomes compared to native populations; this may be due to several factors that include cultural and behavioral patterns, socio-economic determinants, difficulties in accessing healthcare and frequency of pre-existing pathologies [2,3,4].

The definition of syndemic has been proposed for the COVID-19 pandemic, referring to the dependence of population health burden on social determinants and concentration and deleterious interaction of two or more or diseases or other health conditions [5]. The notion of syndemic seems to fit well with the emerging evidence on inequalities in vulnerability, susceptibility, exposure, and transmission of the infection [6]. Furthermore, the COVID-19 burden is unevenly distributed: marginalized populations such as non-Italians and refugees have been impaired by a higher COVID-19 incidence and mortality rate [7]. In western countries, ethnic disparities played a dramatic role during the pandemic: ethnic minorities faced an excess of risks of testing positive for SARS-CoV-2, hospitalization, and adverse COVID-19 health outcomes compared with the White population [8,9,10,11,12,13,14,15,16].

Italian studies about COVID-19 and non-Italians produced a similar but not identical picture. Studies from the early stages of the pandemic found that, compared to Italian cases, non-Italians or people born in a different country were diagnosed later and were more likely to be hospitalized, admitted to ICU, and had a higher risk of death, with differences being more pronounced in those coming from countries with lower Human Development Index (HDI) and in Latinx individuals [17,18,19,20]. A study conducted by the Italian National Institute of Health highlighted a younger age of death in the migrant population compared to the natives [21].

Several studies tried to evaluate whether socio-economic factors mediate this association of ethnicity with COVID-19 incidence and outcomes. A study conducted in Sweden showed that migrants’ country of origin could be associated with their health outcomes: being a non-Italian from a low- or middle-income country was a predictor for a higher risk of death from COVID-19 [22]. In a community-based study from California (USA), low socio-economic status was associated with increased risk of SARS-CoV-2 infection: individuals living in poverty with lower average annual household income, lower rates of employment, or lower rates of health insurance were more likely to test positive [23]. The impact of the area of residence on health outcomes is also confirmed by other studies. For example, in Barcelona, the neighborhoods belonging to the lowest quintile of income had a 42% more incidence of COVID-19 than those belonging to the highest quintile [24]. In New York, more tests were performed in areas with a higher proportion of white residents; conversely, the highest proportion of positive tests was recorded in non-white neighborhoods (especially Black and Latino areas) and in areas defined by a lower socio-economic status [25,26]. In Baltimore, patients living in the most disadvantaged areas, who were predominantly black, had a higher risk for SARS-CoV-2 infection than patients living in less socio-economically deprived areas [27]. 

Cultural norms influencing hygiene practices, social distancing, and information distribution have shaped communities’ pandemic response [28,29]. Moreover, significant disparities have been shown between different ethnicities on the access to information about SARS-CoV-2. Latinx COVID-19 survivors after hospitalization enrolled in a qualitative study reported that, in their experience, COVID-19 was perceived as a distant and secondary threat due to a false feeling of invincibility and misinformation [30]. According to a study conducted in the US, Afro-American and Hispanic citizens had lower knowledge of potential fomite spread, of COVID symptoms and preventive behaviors compared to white people [31]. In fact, non-Italians reported critical barriers to COVID-19-related healthcare services, especially the linguistic barrier: this was particularly evident in research on the Hispanic community of Dallas (Texas), which interpreted the associations between ethnicity and COVID-19 clinical outcomes because of elevated disease severity at admission and limited access to healthcare, especially non-English-speaking Hispanics [32]. 

Further investigation into the socio-economic factors underlying ethnic disparities in COVID-19 contagion and severity requiring hospitalization is needed.

The aim of the study is to analyze differences between Italians and foreign-born people in the risk of testing positive and being hospitalized due to COVID-19 in the population of Umbria, a Region of Central Italy, using a historical cohort design.

## 2. Materials and Methods

Our study was conducted in Umbria, a central Italian region. We considered the population assisted by the health service (906,463 people, which includes domiciled residents and non-residents temporarily present in Umbria). The list of people assisted by the health service is updated half-yearly, excluding those who could no longer be assisted (e.g., dead and emigrants) and including the new ones that could be assisted (e.g., non-Italians). The population file was linked with DBCOVID Umbria, a regional longitudinal database that collects data on SARS-CoV-2 testing results and COVID-19-related outcomes.

During the study period 21 February 2020–31 May 2021 (30 June 2021 for hospitalization follow-up), overall, 54,448 people were positive for SARS-CoV-2, of which 5295 were hospitalized (889 in intensive care) and 1374 died. We decided to end the study period on the 31st of May because of the introduction of self-made antigenic-rapid-tests for SARS-CoV-2 diagnosis and because of the starting of vaccination campaigns.

We considered residence, socio-economic status (SES), rural/urban municipalities index, immigration status (as represented by place of birth), the HDI of birth country, sex, and age groups as individual-level determinants of infection or hospitalization. Municipality of residence was used as a clustering variable. We considered place of birth as a proxy for ethnic and cultural specificities. 

Age groups were defined according to increasing risk of mortality related to SARS-CoV-2 infection (<50, 50–59, 60–69, ≥70 years) [33,34,35]. 

SES was measured at municipalities level by the National Deprivation Index (NDI). To encompass all facets of deprivation in one indicator, the NDI is based on five variables taken from the 2011 Italian population census, including low educational attainment, unemployment, lack of home ownership, one-parent families, and overcrowding.

The HDI is a summary indicator of average performance in three important areas of human development: living a long and healthy life, having access to knowledge, and having a decent standard of living [36]. The life expectancy at birth is used to evaluate the health dimension. The education dimension is calculated using the average number of years of education for adults aged 25 and older and the anticipated number of years of education for children starting school. Gross national income per person is used to measure the standard of living dimension. The HDI uses the logarithm of income to reflect how income becomes less significant as Gross National Income rises. The three HDI dimension indices' scores are then combined using geometric mean to create a composite index. We considered a modified 5 levels classification of HDI index of birth nationality: “Italy” for people born in Italy, “Very high” for HDI higher than Italy, “High” for HDI from 0.7 to Italian HDI, “Medium” for HDI from 0.55 to 0.7 and “Low” for HDI lower than 0.55.

The Urban–Rural Municipalities Index has been proposed by the Italian “Development and Economic Cohesion Department” to categorize municipalities into pole (A), urban (A1), or rural municipalities. Rural municipalities are further classified in three sub-categories based on their distance from the poles, measured in travel time: rural A2, rural B2, and rural C2. Indicators such as the offer of high schools, the presence of a first-level emergency department, and the presence of at least medium/small train stations determine the assignment to the above categories. Descriptive statistics were calculated using frequencies and percentages for categorical variables and mean ± standard deviation (SD) or median and IQR for quantitative variables. The Mann–Whitney Test was performed to compare continuous variables with non-normal distribution. Categorical variables were evaluated by chi-square analysis or Fisher’s exact test were appropriate.

Standardized Cumulative Incidence Rate (SCIR) and Standardized Cumulative Hospitalization Rate (SCHR) for 100,000 inhabitants were calculated using European population 2013. 

We used multilevel logistic regression models to further investigate the influence of country of birth and features of the country of birth on study outcomes (e.g., infection and hospitalization). All patients (level 1) were considered clustering by municipality of residence (level 2). First, we fitted a random intercept empty model (i.e., without fixed effects variables) to test the influence of municipality on response variables. Secondly, a logistic regression model was used to assess individual-level variables as independent determinants. Finally, we fitted a multilevel model including significant individual variables (fixed effects) and allowing the probability of being positive or hospitalized to vary randomly by municipality of residence. 

Models I1 and H1 provided univariate estimations of all covariates considered for infection and hospitalization, respectively.

Multivariate models specification (Model I2–I3 for risk of infection and Model H2–H3 for risk of hospitalization) differ in the presence of the HDI or the geographical macro-area referred to the country of birth.

In all models, Urban–Rural Municipalities Index categories “polo” and “urban” have been merged.

Predicted probabilities at average covariates values were calculated based on models I2, I3, H2, and H3.

A sensitivity analysis was conducted, excluding people under 18 years old and over 70 years of age.

The time elapsed between people’s SARS-CoV-2 positive PCR test and their hospitalization was taken into account: a hospitalization within 48 hours from the diagnosis was used as a proxy for inadequate primary healthcare access.

A logistic regression model was used to evaluate risk factors of hospitalization within 48 hours from the diagnosis (Model H4–H5 for hospitalizations within 48 hours from the diagnosis). Model selection was performed by stepwise backward selection approach (*p* < 0.2).

A *p*-value of less than 0.05 was deemed statistically significant.

Statistical analyses were performed with STATA 16.1 (StataCorpLP, Collage Station TX, USA).

## 3. Results

Among 906,463 individuals included in the analysis, 102,036 were non-Italian nationals (11.3%). The ten most represented foreign nationalities were Romanian (23%), Albanian (15.1%), Moroccan (10.8%), Ukrainian (5.4%), North Macedonian (4.3%), Ecuadorian (2.9%), Nigerian (2.8%), Moldovan (2.5%), Chinese (2.4%), and Philippian (1.9%).

### 3.1. Socio-Demographic Characteristics

Table 1 reports the demographic characteristics of the population.

Non-Italians’ median age, 38 years, was markedly different from Italians’, 51 years. Overall, 72.8% of the non-Italian population was <50 years old, whereas less than 50% (47.8) of the Italian population was younger than 50 years at the time of the study. Women were more represented among foreigners than among Italians (58.4% vs. 52.6%).

A higher percentage of the non-Italian population was in the two more deprived quintiles compared to the Italian population (44.9% vs. 37.4%).

### 3.2. Association of Nationality and Socio-Economic Factors with SARS-CoV-2 Test Positivity 

SCIR for SARS-CoV-2 test positivity was 6539 for Italians and 5753 for non-Italians (Appendix A). The highest SIR among non-Italians was observed in Ecuador (14,327), Albania (7870), Morocco (7307), Nigeria (7098), Ukraine (5329), Moldova (4790), and North Macedonia (4699) populations (Appendix A). The individuals from the three most deprived ID quintiles have higher SCIR of SARS-CoV-2 test positivity (3rd quintile: 6963; 4th quintile: 6933; 5th quintile: 6710 vs. 1st quintile: 5917 and 2nd quintile: 5997) (Appendix A).

Multilevel models are reported in Table 2. Being male (OR 1.05, 95%CI 1.04–1.07), belonging to the most deprived quintile (OR 1.27, 95%CI 1.02–1.59), living in an urban area or in the rural areas closer to the cities (respectively, OR 1.45, 95%CI 1.09–1.93 and OR 1.45, 95%CI 1.21–1.72) and being from Central–South America (OR 1.54, 95%CI 1.43-1.65) were significant risk factors for SARS-CoV-2 test positivity (Model I3). The risk of infection decreased with age (Table 2).

### 3.3. Association of Nationality and Socio-Economic Factors with COVID-19 Hospitalization

Non-Italian individuals have a higher SCHR for COVID-19 (421) than Italians (353) (Appendix A). Among non-Italians, the highest SCHR was for individuals from Nigeria (1799), Ecuador (1204), China (1077), Albania (514), Morocco (479), Philippines (395) and Ukraine (354) (Appendix A). SCHR progressively grows from the highest HDI class (160) to the lowest HDI class (874) (Appendix A). The individuals from the three most deprived ID quintiles have higher SCHR (3rd quintile: 413; 4th quintile: 412; 5th quintile: 387 vs. 1st quintile: 353 and 2nd quintile: 312) (Appendix A).

HDI of the country of origin was a significant predictor of COVID-19 hospitalization (Table 2, Model H2).

Model H3 shows that being male, belonging to the most deprived ID quintile, and living in the rural areas closer to the cities are factors associated with an increased risk of hospitalization (Table 2). Moreover, the risk of hospitalization increases with age (Table 2). With respect to continent of origin, all non-Italian populations faced a significantly increased risk of hospitalization compared to Italians, with people from Central–South America and Africa showing the highest odds ratios (respectively, OR 2.44, 95% CI 1.87–3.18 and OR 1.91, 95% CI 1.56–2.34) (Table 2). People from EU/North America, on the contrary, faced a significantly reduced risk of COVID-19 hospitalization (Table 2). 

Hospitalization within 48 hours from the diagnosis was significantly more likely among non-Italians than among Italians, with a higher risk for people from Africa, Asia, and Oceania (Table 3, Model H5). Hospitalization within 48 hours from the diagnosis was much more likely in people ≥70 years old and medium or low HDI (Table 3, Model H4). 

People from Central–South America have a 1.4% predicted probability of being hospitalized (1.7% for people from the lowest HDI class), against a 0.58% for Italians (Figure 1).

After hospitalization, the probability of accessing the ICU or dying is not different between Italians and foreigners (*p*-value 0.86 and 0.75, respectively).

The sensitivity analysis showed concordant results with all models of our analysis (data not shown).

## 4. Discussion

A population-based study can capture, at best, the impact of COVID-19 on non-Italians compared to the native population and investigate the determinants of the infection and hospitalization trends in these groups. 

We investigated the impact of the COVID-19 pandemic on a relatively large non-Italian community (11.26% of our population). We found that viral circulation assessed through SCIR was lower among non-Italians than among the Italian population. Instead, as shown by Mazzalai et al. [37], disease severity as measured in terms of hospitalization was higher among non-Italians than among Italians. 

The non-Italian population in Umbria was much younger than the Italian-born (median age: 38 vs. 51). Aging indicators for the regional population are among the world’s highest. 

Non-Italians had a lower SCIR for COVID-19 if compared to Italians (5753 vs. 6539). 

Different explanations exist for this observation. Lower use of diagnostic tests can result in an underestimation of the true circulation of the infection. Indeed, a lower rate of diagnostic testing might depend on barriers to primary healthcare access. A systematic review of racial and ethnic disparities regarding COVID-19-related infection, hospitalization, and mortality found that barriers in healthcare access underlie COVID-19-related disparities more than individuals’ comorbid conditions [4]. In Italy, non-Italians have free access to emergency services and many out-patient services, even in case of illegal immigration. However, access to primary care services is limited, and, in particular, receiving the assistance of a self-selected family physician is limited to resident non-Italians that live in Italy with a documented status and other specified categories (e.g., children under the age of 18). Poorer health literacy of non-Italians and lack of tailored and accessible communications from health services and media (e.g., public health campaigns in foreign languages) could also impact non-Italians access to primary care services, including testing [17]. Disparities in testing for SARS-CoV-2 among Italians and non-Italians can also be attributed to differences in testing-related behaviors. For example, given that ethnic minorities are more likely to work in insecure jobs with poor workplace protections, social and economic barriers to testing are probably greater in these communities. Indeed, emerging evidence suggests that people may avoid being tested for fear of losing income or work if necessary for quarantine after testing positive and may be afraid to access official health services due to fear of legal consequences or of repatriation [11,38]. Finally, a percentage of positive tests among non-Italians is lacking because tests carried out abroad were not registered in Italy. 

There was significant heterogeneity among different non-Italian ethnicities in terms of SCIR, which, in some cases, were higher than the Italian population rates. This is confirmed by the multivariate analysis: for example, people from Central and South America showed a high risk of testing positive (OR 1.54, 95%CI 1.43–1.65). People from Ecuador (mean age 35.3 years) had the highest SCIR (14,327). These findings are consistent with published literature regarding the role of ethnicity in COVID-19 outcomes. Several studies have demonstrated a high rate of test positivity among Hispanic people [9,13,39]. A qualitative study describing the experiences of Latinx individuals with COVID-19 reported little use of preventive measures because COVID-19 was perceived as a distant and secondary threat. Both behavioral (e.g., cultural norms) and socio-economic factors (e.g., living in overcrowded houses) impaired the adoption of mitigation measures such as physical distancing [30]. 

People living in the urban areas and in the rural areas close to cities showed an increased risk of SARS-CoV-2 positivity (OR 1.45, 95%CI 1.09–1.93 and 1.21–1.72), which can be linked to high population density, enhanced connectivity and wider geographic access to testing sites [40].

We also observed a high risk of infection among people living in deprived areas (OR 1.27, 95%CI 1.02–1.59). The prevalence of people living in crowded houses, which is more frequent among the deprived, increases household transmission of infection [41]. Furthermore, those living in deprived areas may be at higher risk of occupational transmission since they often have poor working conditions and are less likely to work remotely [41]. In a Belgian study, the incidence in the most deprived areas was 24% higher than in the least deprived areas [42]. According to Khanna et al., patients who were predominantly black and resided in areas with high levels of deprivation had a higher risk of developing COVID-19 than patients who were predominantly white and resided in areas with low levels of deprivation, and this finding reflects health, income, and educational inequities [27]. 

Overall, non-Italians showed a higher SCHR than Italians in our study (421 vs. 353) (Appendix A). Moreover, the SCHR increased with decreasing HDI of the country of birth (p trend<0.01): this is confirmed by the multivariate analysis. People from the lowest HDI class showed a 1.7% predicted probability of being hospitalized, which is higher than that observed for Italian people (i.e., 0.58%) (Figure 1) and a risk of hospitalization within 48 hours from the diagnosis 11 times higher than the Italians (OR 10.9, 95%CI 4.19–28.28) (Table 3).

This observation is in agreement with a recent national study [17]. People from Central–South America, besides having the highest risk of infection, also have the highest risk of being hospitalized for COVID-19 (1.40% predicted probability of being hospitalized). Misinterpretations of the early signs of COVID-19 due to low health literacy, resistance to seeking medical care due to the fear of losing their job, and reduced access to healthcare were described in a qualitative study on the Hispanic community and SARS-CoV-2 infection [30].

In our study, individuals belonging to the most deprived ID quintile had a higher risk of infection and hospitalization (OR 1.27, 95%CI 1.02–1.59 and OR 1.38, 95%CI 1.07–1.78).

Similarly, other studies have found higher hospitalization rates in the most deprived areas of the USA, UK, and France [43,44,45,46] and a small meta-analysis [47] assessing the influence of deprivation on the risk of hospital admission.

The increased risk of infection among individuals living in deprived areas has been linked to many factors, including living in overcrowded homes, lack of or reduced use of personal protective equipment, and barriers to accessing healthcare [45]. 

Noteworthy, the ID level adjustment has little or no influence on health outcomes for non-Italian people in our study.

Mateo-Urdiales et al. instead did not find an association between area-level deprivation and risk of hospitalization from COVID-19 [41]. 

People living in rural areas close to cities faced a higher risk of being hospitalized for COVID-19 (OR 1.25, 95%CI 1.01–1.53). This is probably because the population living in rural areas has, on average, access to fewer physicians and fewer healthcare facilities than the urban population but is subject to higher viral circulation in the case of proximity to cities [40]. 

In our study, males showed a higher risk of being hospitalized for COVID-19 than females (OR 1.58, 95%CI 1.50–1.67). A Danish study found that men with SARS-CoV-2 infection have >50% higher risk of all-cause death, severe COVID-19 infection, or ICU admission than women [48]. A meta-analysis of 3,111,714 cases showed that, although there is no difference in the proportion of males and females with confirmed COVID-19, male patients have a higher risk of ICU admission and death compared with females [49]. 

Males with confirmed COVID-19 had twice the odds of hospital admission compared with females [43]. 

Non-Italians in our study have higher SCHR (421 vs. 351), an increased risk of hospital access, and a markedly increased risk of being diagnosed within 48 hours from hospital admission (Table 3). This observation reinforces the hypothesis of reduced access to primary care, leading to seeking care directly from hospital emergency departments [17]. Moreover, reduced testing and access to primary care can be responsible for the late start of treatments and a late diagnosis in case of unfavorable evolution of COVID 19 disease Thus, our data provide direct evidence for the existence of the diagnostic delay among non-Italians that Fabiani et al. hypothesized in their analysis of national data [17]. 

Other reasons that may lead to inappropriate access to healthcare services include fear of discrimination, poor education, and lack of knowledge about the local health system. Furthermore, emergency services require fewer administrative steps to access, which can reduce language, cultural, and legal barriers [50]. 

Age- and sex-adjusted mortality and ICU rates did not differ between Italians and non-Italians. However, these outcomes were rare in non-elderly people and, consequently, in the non-Italian population. Similar to our study, the study by Fabiani et al., analyzing data from the National COVID-19 surveillance system, reported a lower incidence of SARS-CoV-2 infection and a higher risk of hospital admission among non-Italians if compared to Italians [19]. However, in this study, non-Italians were also more likely to be admitted to intensive care units than Italians, and non-Italians from countries with low HDI faced an increased risk of death [19]. 

Other studies also confirm substantial equality of mortality rates [51,52,53] due to the small sample size [51] and the low overall mortality rates reported by the study [52].

Another Italian study found a significantly higher mortality rate among non-Italians from Latin America than among non-Italians from Asia, Africa, or Central/Eastern Europe [20]. Velasco et al. described that Hispanic inpatients for COVID-19 experienced higher ICU utilization and higher mortality than non-Hispanic patients. This finding was attributed to limited access to healthcare and more severe disease at admission for Hispanic patients, especially in the case of non-English-speaking Hispanics [32]. A phenomenon that could have determined an underestimation of the non-Italian mortality rate is the so-called “salmon bias effect”. When people anticipate passing away soon, they often travel back to their country of origin, but their deaths are not recorded in the statistics of the country where they are currently living [54]. However, this bias is somewhat unlikely in case of acute infection during a pandemic. 

Study limitations. Our study has limitations. In our study population, we could have underestimated the number of non-Italians not legally resident, missing only the ones that had no contact with the National Health Services. A further limitation is that it was not possible to take into account how long non-Italians had been in Italy, even if we did not expect relevant changes in the population due to the traveling restriction active during the study period. Data on comorbidities that can affect the clinical outcomes of COVID-19 (e.g.,: hypertension, obesity, diabetes, cancer, cardiovascular disease, leukemia, etc.) were unavailable. Data on patients’ access to primary healthcare were also lacking. Moreover, we were unable to consider the impact of health literacy and language barriers on study outcomes since this information was not available. Due to the regional coverage and the young age of the non-Italian population, our study was underpowered to detect differences in infrequent health outcomes, including ICU admission and death.

## 5. Conclusions

In conclusion, we found evidence of an increased risk of hospitalization and late diagnosis among non-Italians compared to the Italian-born population. In apparent contrast with high hospital access, the incidence of infection, as measured by test positivity, was lower for non-Italians than for Italians. Indeed, the SCIR showed marked heterogeneity by ethnicity and HDI of the country of origin of non-Italians. Thus, low SCIR could be partly apparent due to reduced access to tests and primary healthcare and missing data on tests performed abroad, and it is partly true as a consequence of reduced contact rates for some ethnic groups, particularly in rural areas.

## Figures and Tables

**Figure 1 ijerph-20-06521-f001:**
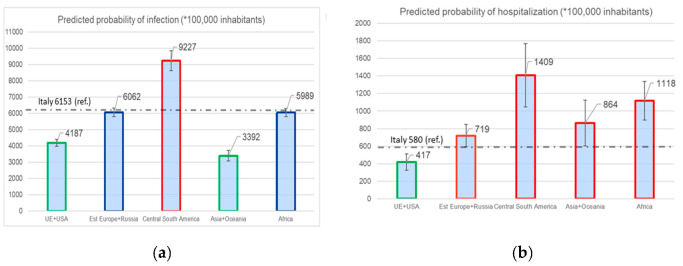
Red outline = significantly higher than Italy; green = s ignificantly lower than Italy; blue = not significantly different from Italy. (**a**) Predicted probability of infection (*100,000 inhabitants); (**b**) Predicted probability of hospitalization (*100,000 inhabitants).

**Table 1 ijerph-20-06521-t001:** Demographic characteristics of a 906,463 people population for COVID-19 cases and hospitalization.

Variables	General Population			COVID-19 Cases				Hospitalization			
	Italian	Non-Italian	Total	Italian	Non-Italian	Total	*p*-Value	Italian	Non-Italian	Total	*p*-Value
	N *(Column%)*	N *(Column%)*	N *(Column%)*	N *(Column%)*	N *(Column%)*	N *(Column%)*		N *(Column%)*	N *(Column%)*	N *(Column%)*	
Sex											
Male	381,131 *(47.4)*	42,402 *(41.6)*	423,533 *(46.7)*	23,843 *(49.2)*	2572 *(43.3)*	26,415 *(48.5)*	<0.001	2755 *(56.3)*	183 *(45.6)*	2938 *(55.5)*	<0.001
Females	423,296 *(52.6)*	59,634 *(58.4)*	482,930 *(53.3)*	24,663 *(50.8)*	3370 *(56.7)*	28,033 *(51.5)*		2139 *(43.7)*	218 *(54.4)*	2357 *(44.5)*	
Age (years)											
<50	384,645 *(47.8)*	74,308 *(72.8)*	458,953 *(50.6)*	26,687 *(55.1)*	4597 *(77.4)*	31,284 *(57.4)*	<0.001	566 *(11.6)*	192 *(47.9)*	758 *(14.3)*	<0.001
50–59	124,723 *(15.5)*	15,320 *(15)*	140,043 *(15.4)*	8111 *(16.7)*	817 *(13.7)*	8928 *(16.4)*		648 *(13.2)*	96 *(23.9)*	744 *(14)*	
60–69	107,223 *(13.3)*	8433 *(8.3)*	115,656 *(12.8)*	5548 *(11.4)*	379 *(6.4)*	5927 *(10.9)*		932 *(19)*	62 *(15.5)*	994 *(18.8)*	
70+	187,836 *(23.4)*	3975 *(3.9)*	191,811 *(21.2)*	8160 *(16.8)*	149 *(2.5)*	8309 *(15.3)*		2748 *(56.2)*	51 *(12.7)*	2799 *(52.9)*	
Geographical areas											
Italy	804,427 (*100*)	-	804,427 *(88.7)*	48,506 (*100*)	-	48,506 (*89.1*)	<0.05	4894 *(100)*	-	4894 *(92.4)*	<0.05
EU and North America	-	31,905 *(31.3)*	31,905 *(3.5)*	-	1389 *(23.4)*	1389 *(2.6)*		-	78 *(19.5)*	78 *(1.5)*	
Central–Eastern Europe	-	29,780 *(29.2)*	29,780 *(3.3)*	-	1919 *(32.3)*	1919 *(3.5)*		-	122 *(30.4)*	122 *(2.3)*	
Central–South America	-	8001 *(7.8)*	8001 *(0.9)*	-	811 *(13.6)*	811 *(1.5)*		-	57 *(14.2)*	57 *(1.1)*	
Asia and Oceania	-	10,904 *(10.7)*	10,904 *(1.2)*	-	408 *(6.9)*	408 *(0.7)*		-	43 *(10.7)*	43 *(0.8)*	
Africa	-	21,446 *(21)*	21,446 *(2.4)*	-	1415 *(23.8)*	1415 *(2.6)*		-	101 *(25.2)*	101 *(1.9)*	
HDI											
Very high	-	5309 *(5.2)*	5309 *(0.6)*	-	138 *(2.3)*	138 *(0.3)*	<0.001	-	14 *(3.5)*	14 *(0.3)*	<0.001
Italy	804,427 *(100)*	-	804,427 *(88.8)*	48,506 *(100)*	-	48,506 *(89.1)*		4894 *(100)*	-	4894 *(92.4)*	
High	-	73,898 *(72.4)*	73,898 *(8.1)*	-	4310 *(72.6)*	4310 *(7.9)*		-	281 *(70.1)*	281 *(5.3)*	
Medium	-	16,561 *(16.2)*	16,561 *(1.8)*	-	1113 *(18.7)*	1113 *(2)*		-	70 *(17.4)*	70 *(1.3)*	
Low	-	6252 *(6.2)*	6252 *(0.7)*	-	380 *(6.4)*	380 *(0.7)*		-	36 *(9)*	36 *(0.7)*	
Missing	-	16 *(0.0)*	16 *(0.0)*	-	1 *(0.0)*	1 *(0.0)*		-	-	-	
Deprivation Index											
Quintile 1	163,933 *(2.4)*	15,995 *(15.7)*	179,928 *(19.8)*	9039 *(18.6)*	789 *(13.3)*	9828 *(18.1)*	<0.001	891 *(18.2)*	54 *(13.5)*	945 *(17.8)*	<0.001
Quintile 2	157,314 *(19.6)*	16,368 *(16)*	173,682 *(19.2)*	8829 *(18.2)*	748 *(12.6)*	9577 *(17.6)*		865 *(17.7)*	41 *(10.2)*	906 *(17.1)*	
Quintile 3	177,664 *(22.1)*	22,205 *(21.8)*	199,869 *(22.1)*	11,629 *(24)*	1385 *(23.3)*	13,014 *(23.9)*		1206 *(24.6)*	97 *(24.2)*	1303 *(24.6)*	
Quintile 4	46,539 *(18.2)*	23,233 *(22.8)*	169,772 *(18.7)*	9204 *(19)*	1856 *(31.2)*	11,060 *(20.3)*		934 *(19.1)*	129 *(32.2)*	1063 *(20.1)*	
Quintile 5	151,600 *(18.8)*	21,336 *(20.9)*	172,936 *(19.1)*	9750 *(20.1)*	1140 *(19.2)*	10,890 *(20)*		992 *(20.3)*	77 *(19.2)*	1069 *(20.2)*	
Missing	7377 *(0.9)*	2899 *(2.8)*	10,276 *(1.1)*	55 *(0.1)*	24 *(0.4)*	79 *(0.1)*		6 *(0.1)*	3 *(0.7)*	9 *(0.2)*	
Urban–Rural Index											
A-pole	366,608 *(45.6)*	51,798 *(50.8)*	418,406 *(46.2)*	23,148 *(47.7)*	24 *(0.4)*	23,172 *(42.6)*	<0.001	2386 *(48.8)*	242 *(60.4)*	2628 *(49.6)*	<0.001
A1-urban	39,676 *(4.9)*	4619 *(4.6)*	44,295 *(4.9)*	3009 *(6.2)*	3459 *(58.2)*	6468 *(11.9)*		274 *(5.6)*	27 *(6.7)*	301 *(5.7)*	
A2-rural	125,119 *(15.6)*	13,687 *(13.4)*	138,806 *(15.3)*	7977 *(16.5)*	350 *(5.9)*	8327 *(15.3)*		776 *(15.8)*	47 *(11.7)*	823 *(15.5)*	
B2-rural	157,942 *(19.6)*	19,115 *(18.7)*	177,057 *(19.5)*	8778 *(18.1)*	806 *(13.6)*	9584 *(17.6)*		881 *(18)*	52 *(13)*	933 *(17.6)*	
C2-rural	107,705 *(13.4)*	9918 *(9.7)*	117,623 *(13)*	5539 *(11.4)*	893 *(15)*	6432 *(11.8)*		571 *(11.7)*	30 *(7.5)*	601 *(11.4)*	
Missing	7377 *(0.9)*	2899 *(2.8)*	10,276 *(1.1)*	55 *(0.1)*	410 *(6.9)*	465 *(0.8)*		6 *(0.1)*	3 *(0.7)*	9 *(0.2)*	
Total	804,427 *(100)*	102,036 *(100)*	906,463 *(100)*	48,506 *(100)*	5942 *(100)*	54,448 *(100)*		4894 *(100)*	401 *(100)*	5295 *(100)*	

%column in italics.

**Table 2 ijerph-20-06521-t002:** Multilevel logistic regression models for COVID-19 cases and hospitalization.

Variables	Covid-19 Cases			Hospitalization		
	Model I1 Univariate	Model I2 HDI	Model I3 Area	Model H1 Univariate	Model H2 HDI	Model H3 Area
	OR (95% CI)	OR (95% CI)	OR (95% CI)	OR (95% CI)	OR (95% CI)	OR (95% CI)
Sex						
Male	1.08 ***	1.05 ***	1.05 ***	1.43 ***	1.58 ***	1.58 ***
	(1.06–1.10)	(1.03–1.07)	(1.04–1.07)	(1.35–1.51)	(1.50–1.67)	(1.50–1.67)
Females	ref.	ref.	ref.	ref.	ref.	ref.
Age (years)						
<50	ref.	ref.	ref.	ref.	ref.	ref.
50–59	0.93 ***	0.93 ***	0.93 ***	3.25 ***	3.40 ***	3.39 ***
	(0.91–0.95)	(0.91–0.95)	(0.91–0.95)	2.93–3.59	(3.07–3.77)	(3.06–3.76)
60–69	0.74 ***	0.74 ***	0.74 ***	5.30 ***	5.69 ***	5.66 ***
	(0.72–0.76)	(0.72–0.76)	(0.72–0.76)	(4.82–5.82)	(5.16–6.26)	(5.14–6.23)
70+	0.62 ***	0.62 ***	0.61 ***	9.06 ***	9.98 ***	9.90 ***
	(0.61–0.64)	(0.60–0.63)	(0.60–0.63)	(8.35–9.82)	(9.18–10.86)	(9.11–10.76)
Nationality						
Italian	ref.			ref.		
Non-Italian	0.97 **			0.64 ***		
	(0.94–0.99)			(0.58–0.71)		
Geographical areas						
Italy	ref.		ref.	ref.		ref.
EU and North America	0.73 ***		0.68 ***	0.41 ***		0.73 ***
	(0.69–0.78)		(0.64–0.72)	(0.33–0.51)		(0.58–0.91)
Central–Eastern Europe	1.07 ***		1.39	0.67 ***		1.23 **
	(1.02–1.13)		(0.94–1.03)	(0.56–0.80)		(1.03–1.48)
Central–South America	1.70 ***		1.54 ***	1.14		2.44 ***
	(1.58–1.83)		(1.43–1.65)	(0.88–1.48)		(1.87–3.18)
Asia and Oceania	0.62 ***		0.55 ***	0.63 ***		1.53 ***
	(0.56–0.68)		(0.50–0.61)	(0.46–0.86)		(1.12–2.09)
Africa	1.08 ***		0.96	0.76 ***		1.91 ***
	(1.02–1.14)		(0.91–1.01)	(0.62–0.92)		(1.56–2.34)
Urban–Rural Index						
A-pole	1.67 ***	1.45 **	1.45 **	1.33 **	1.33	1.32
	(1.29–2.16)	(1.08–1.94)	(1.09–1.93)	(1.05–1.68)	(0.98–1.79)	(0.98–1.79)
A2-rural	1.53 ***	1.44 ***	1.45 ***	1.18	1.25 **	1.25 **
	(1.28–1.84)	(1.21–1.72)	(1.21–1.72)	(0.97–1.43)	(1.01–1.53)	(1.01–1.53)
B2-rural	1.21 **	1.15	1.15	0.98	0.99	0.99
	(1.03–1.43)	(0.98–1.35)	(0.98–1.35)	(0.82–1.19)	(0.81–1.21)	(0.81–1.21)
C2-rural	ref.	ref.	ref.	ref	ref.	ref.
Deprivation Index						
Quintile 1	ref.	ref.	ref.	ref.	ref.	ref.
Quintile 2	1.14	1.11	1.11	1.11	1.17	1.17
	(0.96–1.36)	(0.95–1.30)	(0.95–1.30)	(0.92–1.34)	(0.97–1.42)	(0.97–1.42)
Quintile 3	1.35 ***	1.18	1.18	1.22	1.19	1.20
	(1.09–1.67)	(0.97–1.44)	(0.97–1.45)	(0.99–1.49)	(0.95–1.50)	(0.96–1.50)
Quintile 4	1.48	1.14	1.14	1.34	1.19	1.19
	(0.76–2.87)	(0.60–2.17)	(0.60–2.15)	(0.80–2.24)	(0.65–2.18)	(0.65–2.16)
Quintile 5	1.40 ***	1.27 **	1.27 **	1.34 **	1.37 **	1.38 ***
	(1.11–1.78)	(1.02–1.59)	(1.02–1.59)	(1.07–1.69)	(1.07–1.77)	(1.07–1.78)
HDI						
very high	0.44 ***	0.45 ***		0.46 ***	0.47 ***	
	(0.37–0.53)	(0.38–0.53)		(0.27–0.79)	(0.28–0.79)	
Italy	ref.	ref.		ref.	ref.	
high	1.37	0.88 ***		0.62 ***	1.25 ***	
	(0.94–1.00)	(0.85–0.91)		(0.55–0.70)	(1.10–1.41)	
medium	1.11 ***	0.99		0.68 ***	1.65 ***	
	(1.04–1.18)	(0.93–1.05)		(0.53–0.86)	(1.30–2.10)	
low	1.20	0.89 **		0.94	3.05 ***	
	(0.92–1.13)	(0.80–0.89)		(0.67–1.31)	(2.17–4.28)	
σ^2^u		0.086	0.085		0.07	0.07

** *p* < 0.05; *** *p* < 0.01

**Table 3 ijerph-20-06521-t003:** Logistic regression models for risk factors of hospitalization within 48 hours from the diagnosis.

Variables	Hospitalization within 48 Hours from the Diagnosis	
	Model H4	Model H5
	OR (95% CI)	OR (95% CI)
Age (years)		
<50	ref.	ref.
50–59	0.94	0.94
	(0.76–1.17)	(0.76–1.16)
60–69	1.36	0.96
	(0.78–1.17)	(0.78–1.17)
70+	1.8 ***	1.80 ***
	(1.51–2.14)	(1.51–2.14)
Geographical areas		
Italy		ref.
EU (North America)		2.12 ***
		(1.34–3.45)
Central–Eastern Europe		1.89 ***
		(1.31–2.72)
Central–South America		1.93 **
		(1.14–3.28)
Asia and Oceania		4.5 ***
		(2.28–8.87)
Africa		4.58 ***
		(2.94–7.12)
HDI		
very high	1.8	
	(0.61–5.25)	
Italy	ref.	
high	2.05 ***	
	(1.6–2.64)	
medium	4.52 ***	
	(2.67–7.66)	
low	10.9 ***	
	(4.19–28.28)	

** *p* < 0.05; *** *p* < 0.01.

## Data Availability

The datasets generated and analyzed during this study are available, at any time, upon request at alessio.gili@unipg.it.

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
