# Peer review of "Socio-Economic Factors Associated with Ethnic Disparities in SARS-CoV-2 Infection and Hospitalization"

_ijerph, 2023, doi:10.3390/ijerph20156521_

Round 1

Reviewer 1 Report

I read the proposed manuscript with great interest. The authors associated socio-demographic factors with ethnic disparities in SARS-CoV-2 infection and hospitalization. They concluded that the risk of hospital admission was higher among non-Italians. The article is well-written and the statistical approach is appropriate. However, there are some aspects that should be improved.

- The abstract lacks the numerical results for the non-Italian population

- Better to avoid the term "racial" line 66

- Did the authors consider the 48-hour cutoff either as a proxy of inadequate primary health care access or as a proxy for contextual hospitalization at diagnosis? This aspect is not very clear

- The work has many limitations of which the authors are aware. A further limitation not taken into account is how long the non-Italians had been in Italy. this variable has a major impact on all the results.

- The references are not all edited according to the standards for authors

- The authors have not reported anything in the ethical approval section. If necessary it should be included.

Author Response

REVIEWER N°1

1. The abstract lacks the numerical results for the non-Italian population

Thank you for your comment. The percentage written in the abstract (9.7% hospitalizations) was related to the total number of hospitalizations of both Italians and non-Italians. Therefore, we corrected that sentence in the abstract.

2. Better to avoid the term "racial" line 66 

Thank you for your suggestions. Even if we found that terminology in much of the literature we consulted, we tried to avoid using that word in our manuscript. We missed that sentence, and we now changed into: “higher proportion of white residents

3. Did the authors consider the 48-hour cutoff either as a proxy of inadequate primary health care access or as a proxy for contextual hospitalization at diagnosis? This aspect is not very clear

We thank the Reviewer for having highlighted this point. The 48-hour describes contextual hospitalizations at diagnosis. We considered contextual hospitalization at diagnosis as a proxy of inadequate primary health care access. We canceled the sentence: “The time of 48 hours between positive test and hospitalization was considered as a proxy for a contextual hospitalization and diagnosis” and left the more accurate sentence: “The time elapsed between people’s COVID-19 positive PCR test and their hospitalization was taken into account: a hospitalization within 48 hours from the diagnosis was used as a proxy of inadequate primary health care access”  in the methods to clarify this issue.

4. The work has many limitations of which the authors are aware. A further limitation not taken into account is how long the non-Italians had been in Italy. this variable has a major impact on all the results.

We appreciate this consideration and understand the importance of this issue. We thought about this while we designed the study: since there were traveling restrictions due to the COVID-19 pandemic, we expect that the population of both Italians and not-Italians did not have relevant changings during the study period. However, in the submitted draft of the discussion, we took into account the tests that could have been taken abroad: “Finally, a percentage of positive tests among non-Italians is lacking, because tests carried out abroad were not registered in Italy”. We added this sentence to the limitations paragraph: “A further limitation is that it was not possible to take into account how long non-Italians had been in Italy, even if we did not expect relevant changes in the population due to the traveling restriction active during the study period.”

5. The references are not all edited according to the standards for authors 

Thank you, we edited the references according to the standards for authors.

6. The authors have not reported anything in the ethical approval section. If necessary it should be included.

We received an Ethic Committee approval from the Regional Ethic Committee of Umbria, Italy "CER Umbria" (CER N 4183/19), prot.n. 23155/21/ON, on the 27/10/2021. We added the reference in the manuscript.

https://www.ospedale.perugia.it/strutture/comitato-etico-regionale-cer-umbria

Reviewer 2 Report

This article presents inequality between Italian and non-Italian residents, in relation to incidence and hospitalization of COVID-19. It is an interesting article which focus on identifying the specific groups at higher risk of incidence and hospitalization for COVID-19.

I have some comments which might direct the improvement of understanding some of the results.

1.   It is clear from the distribution presented in table 1, that there are two sub-population. However, the presentation of the distributions of cases and hospitalized people is not informative enough. The presentation should have been used specific rates for each cell in the table (e.g., % of COVID-19 cases in each age and ethnic group).  

2.   The methodology section should give much more details, which will help understanding the results:

a.       One of the main independent variables is the HMI. The authors could assist the readers by referring to a reference (or a website) which explain this variable and presents the data.

b.       The Urban-Rural Municipalities variable refer to several subcategories of rural area (A2, B2, C2) without any explanation. It appears in the tables, but the reader can only guess which code refer to which distance from the city.

c.        Table 3 refer to models H4 and H5 which are not mentioned at all in the methodology.

3.   In the results, it is mentioned that " SCIR for COVID-19 test positivity" showed higher rates among Italians, than non-Italian, without further details. From table 2 we can learn that these were the results of a univariate analysis. Multivariate analysis present more specific results that can then direct the discussion, but it is not written clearly enough in the results.

4.   In the discussion, the authors refer to the results, one by one, instead of integrating it and explain the overall findings. For example, they started with explaining why non-Italians had a lower SCIR for SARS-CoV-2 compared to Italians. In the multivariate analysis they could focus their finding by showing the heterogeneity of the non-Italians and the association of SCIR with their country of origin. The discussion should have focus on the multivariate findings (which also controlled for age and sex).  

5.   In the discussion the authors claim that "People living in the rural areas close to cities, faced a higher risk of being hospitalized for COVID-19 (OR 1.25, 95%CI 1.01-1.53). This is probably because, population living in rural areas is on average older…". As the multivariate analysis controlled for age – this is not a legitimate explanation.

6.   On line 296 it is written – "Moreover, the SCHR increased with the HDI of the country of birth" in fact, in table 2 we see the opposite.

Minor remarks:

1.   The authors use SARS CoV 2 and COVID-19 alternately, instead of being consistent with one of the expressions.

2.   In row 112 there is a definition of age groups. " (<50, 50-59, 60-69, >70 years)". The last category should have been 70. Otherwise, the scale does not include the 70 years old people. It appears again as >70 in the text in row 221. In tables 1-3 it is written correctly (70+).

3.   In row 331 there is an expression ITU. It seems to me that it is a typing mistake, and it should be ICU.

Author Response

REVIEWER N°2

  1. It is clear from the distribution presented in table 1, that there are two sub-population. However, the presentation of the distributions of cases and hospitalized people is not informative enough. The presentation should have been used specific rates for each cell in the table (e.g., % of COVID-19 cases in each age and ethnic group).

Thank you for your comment. We added the Geographical Areas of origin in Table1 as they are presented in Table2. The percentages of COVID-19 cases and hospitalizations are written in brackets in italics.

  1. The methodology section should give much more details, which will help understanding the results.

a. One of the main independent variables is the HMI. The authors could assist the readers by referring to a reference (or a website) which explain this variable and presents the data.

We thank the Reviewer. We added a reference about the Human Development Index (HDI) in the methods section.

 b. The Urban-Rural Municipalities variable refer to several subcategories of rural area (A2, B2, C2) without any explanation. It appears in the tables, but the reader can only guess which code refer to which distance from the city.

Thank you for highlighting this issue. We added a sentence regarding the code for the Urban-Rural Municipalities in the methods section: “ The Urban-Rural Municipalities Index has been proposed by the Italian “Development and Economic Cohesion Department” to categorize municipalities into pole (A), urban (A1) or rural municipalities. Indicators such as the offer of high schools, the presence of a first-level emergency department and the presence of at least medium/small train stations, determine the assignment to the above categories. Rural municipalities are further classified in three sub-categories based on their distance from the poles, measured in travel time: rural A2, rural B2, rural C2”.

c. Table 3 refer to models H4 and H5 which are not mentioned at all in the methodology.  

Thank you for your comment, we added a reference of Models H4 and H5 in the methods section.

  1. In the results, it is mentioned that " SCIR for COVID-19 test positivity" showed higher rates among Italians, than non-Italian, without further details. From table 2 we can learn that these were the results of a univariate analysis. Multivariate analysis present more specific results that can then direct the discussion, but it is not written clearly enough in the results.

We thank the Reviewer for this comment. We want to underline that the SCIR data is in Table S1 and represents the Standardized Cumulative Incidence Rate for 100,000 inhabitants (European population 2013), not the result of a regression analysis. On the other hand, Table 2 presents the multilevel logistic regression models for COVID-19 cases. In order to underline the relevance of the multivariate analysis, we reported in the text the Odds Ratio (already presented in Table2) of COVID-19 positivity for the individuals from Central-South America. We did the same for the Odds Ratio of hospitalization for the individuals from Central-South America and Africa.

  1. In the discussion, the authors refer to the results, one by one, instead of integrating it and explain the overall findings. For example, they started with explaining why non-Italians had a lower SCIR for SARS-CoV-2 compared to Italians. In the multivariate analysis they could focus their finding by showing the heterogeneity of the non-Italians and the association of SCIR with their country of origin. The discussion should have focus on the multivariate findings (which also controlled for age and sex). 

We thank the Reviewer for this suggestion. We highlighted in the discussion that the SCIR results were confirmed by the multivariate analysis : “There was a significant heterogeneity among different non-Italians ethnicities in terms of SCIR, which, in some cases, were higher than the Italian population rates. This is confirmed by the multivariate analysis:F for example, people from Central and South America showed a high risk of testing positive (OR 1.54, 95%CI 1.43-1.65)”. We did the same for hospitalization: “the SCHR increased with decreasing HDI of the country of birth (p trend<0.01): this is confirmed by the multivariate analysis.

  1. In the discussion the authors claim that "People living in the rural areas close to cities, faced a higher risk of being hospitalized for COVID-19 (OR 1.25, 95%CI 1.01-1.53). This is probably because, population living in rural areas is on average older…". As the multivariate analysis controlled for age – this is not a legitimate explanation.

We are grateful to the Reviewer for this comment which gave us the opportunity to correct this. Since the multivariate analysis is controlled for age, we removed “older” from the sentence.

  1. On line 296 it is written – "Moreover, the SCHR increased with the HDI of the country of birth" in fact, in table 2 we see the opposite.

We thank the Reviewer for this input. We want to underline that the SCHR data is in Table S1 and represents the Standardized Cumulative Hospitalization Rate for 100,000 inhabitants (European population 2013). On the other hand, Table 2 presents the multilevel logistic regression models for COVID-19 hospitalization. We corrected that line in the text with this sentence: “Moreover, the SCHR increased with decreasing HDI of the country of birth

Minor remarks:

  1. The authors use SARS CoV 2 and COVID-19 alternately, instead of being consistent with one of the expressions.

Thank you for this comment. We used “SARS-CoV-2” when referring to the pathogen and “COVID-19” when talking about the infectious disease. We corrected the few misuses of these terms in the text.

  1. In row 112 there is a definition of age groups. " (<50, 50-59, 60-69, >70 years)". The last category should have been ≥70. Otherwise, the scale does not include the 70 years old people. It appears again as >70 in the text in row 221. In tables 1-3 it is written correctly (70+).

We appreciate this suggestion. We wrote “≥70” in both parts of the text that you highlighted.

  1. In row 331 there is an expression ITU. It seems to me that it is a typing mistake, and it should be ICU.

Thank you for bringing this to our attention. We corrected the typing mistake.
